# Associations between Adverse Childhood Experiences within the Family Context and In-Person and Online Dating Violence in Adulthood: A Scoping Review

**DOI:** 10.3390/bs12060162

**Published:** 2022-05-24

**Authors:** Raúl Navarro, Elisa Larrañaga, Santiago Yubero, Beatriz Víllora

**Affiliations:** 1Department of Psychology, Faculty of Education and Humanities, University of Castilla-La Mancha, Avda de los Alfares, 42, 16071 Cuenca, Spain; santiago.yubero@uclm.es (S.Y.); beatriz.villora@uclm.es (B.V.); 2Department of Psychology, Faculty of Social Work, University of Castilla-La Mancha, C/ Camino Cañete, s/n, 16071 Cuenca, Spain; elisa.larranaga@uclm.es

**Keywords:** dating violence, intimate partner violence, adverse childhood experiences, scoping review, adulthood

## Abstract

*Background*: Adverse childhood experiences (ACEs) are a common pathway to risky behaviour, violence or re-victimisation, disability, illness, and premature mortality and, as such, may be associated with victimisation and perpetration of dating violence not only in adolescence but also in adulthood. *Method:* A scoping review was performed in accordance with PRISMA guidelines. Four databases (Web of Science, Scopus, PubMed, and PsycINFO) were used to search for studies published between 2000 and 2021 that analysed the relationship between adverse childhood experiences within the family context and the perpetration or victimisation of dating violence in adulthood. *Results*: The search yielded 599 articles, 32 of which met the inclusion criteria and were ultimately included in the review. Most of the study samples were from the United States. Most of the studies sampled university populations. The studies had a clear objective, were of an appropriate design, contained a detailed description of the sample, and used valid and reliable measurement instruments. *Conclusion:* This scoping review shows that the relationship between ACEs and perpetration and/or subsequent victimisation is complex and that, while adverse childhood experiences are a factor associated with adult dating violence, they are likely to coexist with other personal, family, and environmental problems. Therefore, adverse childhood experiences may not be a necessary or sufficient condition for experiencing dating violence.

## 1. Introduction

Intimate partner violence is a major public health issue with multiple implications for mental health [1,2,3]. It occurs not only among married or cohabiting couples, but also among those who are dating or in an affective and/or sexual relationship, regardless of sexual orientation [4,5]. When intimate partner violence takes place between dating partners, it is known as dating violence [6]. Within the scientific community, there is no clear consensus on the concept of dating violence. Stonard, Bowen, Lawrence, and Price [7] define dating violence as any violent, abusive, threatening, controlling, or stalking behaviour directed towards a partner or ex-partner in the context of a dating relationship. Straus [8] defined the dating relationship as a relationship between two people that includes opportunities for social interaction and shared activities, with an explicit or implicit intention to continue the relationship until one of the two parties ends it or some other more committed relationship (e.g., cohabitation or marriage) is established.

Dating violence comprises four types of behaviours [9]: (1) Physical violence: assaulting or trying to harm a partner by hitting, kicking, or using other physical force; (2) sexual violence: forcing or attempting to force a partner to engage in a sexual act, sexual contact, or physical or non-physical activity (e.g., sending photos of a sexual nature and/or demanding unwanted sexual acts through digital media); (3) psychological aggression: using verbal and/or non-verbal communication with the intention of mentally or emotionally harming and/or controlling another person; (4) stalking: a pattern of repeated and unsolicited surveillance and contact by a partner that causes the recipient fear or concern.

As a widespread social problem, dating violence has a significant impact on the health of its victims in several areas. Regardless of sexual orientation, victims of all forms of dating violence (physical, sexual, and psychological violence, and stalking) were found to have lower levels of physical and psychological health [1,10], as well as greater academic difficulties [11]. Various factors put people at high risk of experiencing or perpetrating dating violence. These factors include gender inequality, racial discrimination, homophobia, or poverty [12], negative interpersonal relationships such as “having deviant peers” [13], and negative events during childhood and adolescence [14,15], with particular relevance given to factors such as “witnessing parental violence” [13].

Negative events during childhood can be grouped into what are known as Adverse Childhood Experiences (hereafter ACEs). ACEs were initially defined as child abuse and domestic abuse [16]. Experiences included (but were not conceptually limited to) harms directly affecting children such as abuse (emotional, physical, and sexual) and neglect (physical and emotional). Also included were harms affecting children indirectly through their living environments, such as growing up in homes with domestic violence, household members who abuse alcohol or drugs or have mental disorders, illness, relationship stress (such as separation or divorce), or where household members engage in criminal behaviour. This set of experiences was later expanded to include ACEs from both developing and developed countries, with the addition of collective violence in the community, early compulsory military service, exposure to bullying, other forms of peer violence, and physical and emotional violence between siblings [17]. This review will focus on ACEs that take place within the family context, using the classification developed by Felitti and Anda [18], which includes the categories of child abuse, neglect, and household dysfunction.

A growing body of research has made it increasingly clear that adverse childhood experiences are a critical public health issue [19,20,21]. A nationwide representative study from the USA revealed that 61.55% of the sample had reported at least one ACE and 24.64% had reported three or more [22]. Stressful or traumatic childhood experiences are a common pathway to social, cognitive, and emotional impairments leading to greater risk of unhealthy behaviours, violence or re-victimisation, disability, illness, and premature mortality [17] and, therefore, may be associated with victimisation and perpetration of dating violence.

The emergence of dating violence as a form of intimate partner violence is linked to the onset of dating and sexual experimentation in early adolescence, between the ages of 10 and 14 [23]. As relationships become more serious and stable during adolescence, conflict may escalate with relational dynamics based on domination and aggression starting to emerge [24,25]. Dating violence mainly affects adolescents, but also occurs among adults. In fact, young adults aged 20–24 years old are most at risk of perpetrating and experiencing intimate partner violence [26,27]. Several studies show that prevalence ranges from 23–38% among emerging adults [8,28], declining thereafter until the age of 35 [29], by which time many of these relationships, if not already over, have become cohabiting relationships or have reached marital status.

For the purpose of this review, dating violence is understood as a form of intimate partner violence occurring not only in adolescence, but also between emerging adults, young adults, and older dating partners. The heterogeneous nature of the research on dating violence, due to the use of different definitions, sampling, and data analysis, combined with the fact that there has been less research on dating violence among adults than among adolescents [30], makes it difficult to conduct systematic reviews and meta-analyses on its relationship to adverse childhood experiences. However, a comprehensive synthesis of the available data on this association is required, so a scoping review has been selected. The present scoping review was conducted to explore the associations between ACEs within the family context and dating violence perpetration and victimization in real and online environments in adulthood. Therefore, this scoping review synthesises the literature analysing the relationships between adverse childhood experiences and dating violence among people over the age of 18, both as a means of raising awareness of the importance of negative childhood events for personal and social development and providing recommendations for preventing dating violence.

## 2. Materials and Methods

### 2.1. Design

A scoping review was performed of the published literature on the relationship between adverse childhood experiences within the family context and perpetration or victimisation of intimate partner violence in adulthood. Scoping reviews are an excellent tool for providing an overview of the scientific evidence on a specific topic, examining how research has been conducted on that topic, describing the volume of research, and identifying the main factors related to the topic under study [31]. They have been used to map research on the nature, patterns, and consequences of different forms of intimate partner violence. Malhi et al. [32], for example, used this approach to synthesise the scientific evidence on what influences male perpetration of dating violence during adolescence. Reyes et al. [3] conducted a scoping review to explore the mental health implications of intimate partner violence by past or current romantic partners among Hispanic women in the United States. Afrouz [33] used a scoping review to explore the nature, patterns, and consequences of intimate partner violence perpetrated using technology.

No protocol was registered for this project. The present scoping review followed the five-step methodological framework proposed by Arksey and O’Malley [34]. The screening of the articles included in this review, as well as the summary and reporting of the results obtained, is described in line with the Preferred Reporting Items for Systematic Reviews and Meta-Analyses extension for Scoping Reviews (PRISMA-ScR) [35].

### 2.2. Identification of the Research Question

The research question for this scoping review was: *what adverse childhood experiences within the family context are associated with dating violence in adulthood?* In addition, the aim was to find out which forms of dating violence have been examined by the research and which are most closely related to ACEs. With regard to the latter, an important question was which ACEs have been most and least studied in relation to intimate partner violence. Dating violence has been defined as a form of intimate partner violence that includes any form of physical violence, psychological aggression, sexual violence, stalking, or a combination of one or more of these, between current or former dating partners [36]. In keeping with previous research [37], adulthood encompassed individuals aged 18 and older. Adverse childhood experiences were defined as “childhood events, varying in severity and often chronic, occurring in a child’s family or social environment that cause harm or distress, thereby disrupting the child’s physical or psychological health and development” ([38], p. 1489).

### 2.3. Identification of Studies

Four bibliographic databases were used for this review: Web of Science (WoS), Scopus (SP), PubMed (PM), and PsycINFO (PI). The literature search was conducted in October 2021 and was restricted to English language articles published during or after 2000. It was decided to focus the search on five dimensions with a search string for each one (see Table 1 for all dimensions and terms used): Participants (Dimension 1), Dating Violence (Dimension 2), Roles Played in Dating Violence (Dimension 3), Adverse Childhood Experiences (Dimension 4), and Study Design (Dimension 5). To ensure as complete a search as possible, variants or synonyms of the established search terms were used.

### 2.4. Study Selection

The search was conducted by the first and last authors (RN and BV, respectively). Both also screened and selected the records resulting from each database search, discussing any questionable records until reaching agreement. Any remaining doubts or disagreements were discussed and resolved with the other authors.

The articles were screened using the following criteria (see Table 2): (1) The article had to analyse at least one adverse childhood experience within the family context in accordance with Felitti and Anda’s classification [18]. Studies that did not include at least one of these adverse experiences and their association with dating violence were excluded. (2) Study participants had to be 18 years of age or older. Although dating violence is usually understood as a form of intimate partner violence occurring between the ages of 10 and 24 [6], other studies include samples up to 40 years of age [39] and beyond [40]. For this reason, all studies that specifically looked at dating violence among participants over the age of 18 were included. All studies involving participants under the age of 18 were excluded. (3) Study participants had to be from the general population; therefore, studies involving specific groups (e.g., federal offenders or psychiatric patients) were excluded. (4) Studies had to be published in English, in peer-reviewed journals, and be quantitative. Studies that did not meet these criteria were excluded from the final analysis.

Figure 1 shows the PRISMA flow chart illustrating the identification, screening, and selection process. The search for combinations of terms in the four databases yielded 599 articles. Ninety-five duplicate results from the different databases were eliminated, leaving five hundred and four for the first phase of screening by title and abstract. Three hundred and seventy-nine articles were eliminated in this first screening, leaving one hundred and twenty-five articles for the full screening. A total of 93 articles were eliminated after full screening against the above research question and inclusion criteria, and the reasons for their removal are shown in the flow chart. In the end, 32 articles met the above-mentioned inclusion criteria and were included in the review.

### 2.5. Charting and Data Analysis

Garrad’s matrix method [41] was used to extract information from each of the studies included in the final selection and synthesise their findings. The matrix (see Table A1 in Appendix A) includes the geographic location, year of publication, sample size, sex and age of the participants, study objectives, forms of dating violence analysed, roles played in dating violence (i.e., perpetration, victimisation, or both), instruments used to measure dating violence, type of ACEs examined and the instruments used to do so, type of statistical analysis used to analyse the relationship between dating violence and ACEs, and the main findings on this relationship. The selected articles were organised in chronological order so as to better visualise the evolution of the literature on the relationship between dating violence and ACEs.

### 2.6. Study Rigour

Coauthors EL and SY tested the rigour and quality of the selected studies using the criteria employed by Reyes et al. [3] in their scoping review on intimate partner violence and mental health outcomes. Each of the articles was evaluated using the following criteria: (1) Was the study objective/question clearly stated? (2) Was the study methodology used appropriate for the research question? (3) Was the sampling method concisely described? (4) Could the sample selection process be biased in any way? (5) Was the study sample representative of the general population to which the findings are attributed? (6) Was the sample calculation based on a statistical power analysis? (7) Was an adequate response rate achieved? (8) Are the measurement instruments valid and reliable? (9) Was statistical significance assessed? (10) Do the main findings include confidence intervals? Each question was answered with “yes,” “no,” or “not sure,” using the sum of the “yes” responses to create a quality index (Item 4 responses were reversed). Higher scores indicate higher quality in the conduct of the study.

## 3. Results

### 3.1. Sample Characteristics

Table A1 shows the sample characteristics (see Appendix A). Most of the study samples were from the United States (*n* = 25; 78.1%), the United States and Canada (*n* = 1; 3.1%), Italy (*n* = 1; 3.1%), Greece (*n* = 1; 3.1%), the United Kingdom (*n* = 1; 3.1%), South Korea (*n* = 1; 3.1%), and Australia and New Zealand (*n* = 2; 6.2%). Only one study from the United States and South Korea (*n* = 1; 3.1%) included cross-cultural analysis. No studies were found from Spain or Latin America. The studies included in the review were conducted between 2002 and 2021, of which 53% were carried out between 2017 and 2021.

The studies analysed comprised samples of adults aged 18–61 years old. Most studies included both male and female participants (*n* = 26; 81.2%). Seven of the studies comprised single-sex samples, where only men (*n* = 3; 9.3%) and only women (*n* = 4; 12.5%) were assessed. For the most part, data collection was limited to information provided by only one partner in a relationship; only three studies included couples (specifically, heterosexual couples). No studies involving LGBT participants were found. Of the publications analysed, 25 focused on university populations, six on the general population, and only one came from national child protection registers.

### 3.2. Methodological Differences

Of the 32 international articles included in this scoping review, 24 were of cross-sectional design (75%), 6 were of longitudinal design (18.7%), and 2 were of quasi-experimental design (6.2%).

Among the instruments used to measure dating violence, the Conflict Tactics Scale (CTS) [42], (*n* = 5; 15.6%), the Conflict Tactics Scale Revised [43], (*n* = 2; 6.2%), and the Conflict Tactics Scale 2 (CTS2) [44], (*n* = 17; 53.1%) were the most used (*n* = 26; 81.25%). Instruments used to measure cyber dating abuse included: the Cyber Aggression in Relationships Scale (CARS) [45], (*n* = 1; 3.1%), the Digital Dating Abuse scale (DDA) [46], (*n* = 1; 3.1%), the Cyber Dating Abuse Questionnaire [47], (*n* = 1; 3.1%), and the Partner Cyber Abuse Questionnaire [48], (*n* = 1; 3.1%). ACE measures varied widely although the Conflict Tactics Scale [42,44,49] was again the most used (*n* = 10; 31.2%).

In terms of statistical analyses, 50% of the articles used bivariate analyses, 96.8% multivariate analyses, 21.8% mediation path analyses, 15.6% moderation analyses, 12.5% analysed the cumulative effects of ACEs, and 12.5% included interaction analyses or propensity score analyses. Mediating and moderating variables included: early maladaptive schemas, fearful dating experiences, attachment style, self-control, anger, hostility, communication of emotions, empathy, attitudes to violence, delinquency, and risky behaviours (alcohol use, drug use, and risky sexual behaviour).

### 3.3. Study Rigour Scores

Overall, most of the studies had relatively high quality scores, with the lowest scores being five out of a possible ten points. The criteria used indicated that all studies had a clear study objective, were of appropriate design for the research questions, provided a detailed description of how the sample was obtained, reported statistical significance, and had valid and reliable measurement instruments. Among the main problems with the studies reviewed were possible sample selection bias (56.2%) and failure to mention whether sample selection was made on the basis of a power analysis (75%). Table 3 shows all the rigour scores for each of the studies analysed.

### 3.4. Primary Analyses

The results are organised by the forms of dating violence examined and, for each one, the relationships with the ACEs analysed by the studies included in this scoping review are summarised. Table A1 provides a summary of the main findings of each study on the relationship between forms of dating violence and ACEs.

#### 3.4.1. Findings on Global Dating Violence and ACEs

Seven studies (21.8%) measured dating violence as a global variable as opposed to examining different types of violence. These studies found that the perpetration of dating violence is related to various adverse childhood experiences: physical abuse [24,73], emotional abuse [24,73], sexual abuse [24], physical and emotional neglect [58,73], low parental warmth [58], parental mental illness, parental suicide attempt, parental criminal conviction, and parental separation [24]. However, some of these studies did not report a positive association between perpetration and sexual abuse [58,67], interparental violence, parental substance abuse [24], or violent socialisation in childhood [67].

These same studies reported a positive relationship between dating violence victimisation and adverse childhood experiences such as: physical abuse [24,51,58,64,73], emotional abuse [24,73], sexual abuse [24,39,51,64], physical and emotional neglect [49,67], witnessing interparental violence [24,51], low parental warmth [58], substance abuse, mental illness, suicide attempts, criminal conviction, and parental separation [24]. However, other studies did not support an association between dating violence victimisation and violent socialisation in childhood [67], sexual abuse [67,73], emotional neglect or interparental violence [24].

#### 3.4.2. Physical Dating Violence and ACEs

##### Childhood Physical Abuse

Sixteen studies (50%) examined the relationship between childhood physical abuse and physical dating violence. A positive association was found between being a victim of childhood physical abuse and perpetration of physical dating violence [55,56,61,65,70,72,73,78]. However, Jennings et al. [60] reported that, while victims of childhood physical abuse were significantly more likely to report perpetration of physical dating violence, once adults who had experienced childhood physical abuse were matched with a sample of adults who had not, both groups were equally likely to perpetrate dating violence. Other more recent studies also failed to find statistically significant associations between the two variables [40,69].

Similarly, there was no clear direction in the study findings with regard to dating violence victimisation. Several studies reported a positive association between childhood physical abuse and dating victimisation [53,55,63,70,76]. However, other studies found no such association [40,54,59,69]. Research elsewhere has found that while there is a relationship between childhood physical abuse and physical dating victimisation, this is not a causal relationship. Instead, it occurs when other adversities are experienced within the family context, such as witnessing interparental violence in the home [60].

##### Childhood Psychological/Emotional Abuse

Seven studies (21.8%) examined the relationship between childhood psychological/emotional abuse and physical dating violence. Most of the reviewed research found no significant associations [40,59,72,73]. However, Baller and Lewis [74] found that greater exposure to psychological abuse during childhood was associated with higher perpetration of physical dating violence. By contrast, more studies reported a positive association between childhood psychological/emotional abuse and physical dating victimisation [40,63,74]. However, this is not the case for all the studies reviewed [76].

##### Childhood Sexual Abuse

Seven studies (21.8%) analysed the relationship between childhood sexual abuse and physical dating violence. Most of these found no significant associations with the perpetration of physical dating violence [54,59,73,76], while other studies did [40,62]. There were also mixed results for victimisation. Cross-sectional analyses controlling for other covariates [62] and other forms of child abuse [40] and longitudinal analyses [63] found that victims of childhood sexual abuse are significantly more likely to experience physical violence victimisation. However, other cross-sectional and longitudinal studies did not find the same association [54,73,76].

##### Witnessing Interparental Violence

Nine studies (28.1%) examined the association between witnessing interparental violence during childhood and physical dating violence. Most studies found that interparental violence was positively associated with the perpetration of physical violence [55,66,69,72,76], although not all studies reported this relationship [55,56,60,78]. Similarly, certain studies reported a positive relationship between witnessing interparental violence and physical dating violence victimisation [55,69,76], whereas other research found no such relationship [60,70]. In their cross-cultural study with participants from the United States and South Korea, Gover et al. [56] found a positive association when the father was the perpetrator of domestic violence as opposed to the mother, but only among the US participants.

##### Parental Neglect

Five studies (15.6%) examined the association between parental neglect and physical dating violence. Most studies found a positive association with the perpetration of physical violence [59,69,73]. However, these results are not so clear when looking at the type of neglect (physical or emotional) and the sex of the participants. For example, Dardis et al. [59] found that childhood experience of maternal physical neglect was predictive of physical violence perpetration in men, but not in women. By contrast, Nikulina et al. [76] found no association between physical or emotional neglect and the perpetration of physical violence. With regard to physical dating violence victimisation, most studies reported an association between this and being a victim of childhood neglect [63,69,73]. However, Nikulina et al. [76] did not find the same relationship.

##### Other Adverse Childhood Experiences

Six studies (18.7%) examined other adverse childhood experiences in relation to physical dating violence. Perpetration was positively associated with experience of abuse in dysfunctional homes [74], the quality of the maternal relationship [70,78], and growing up with an incarcerated caregiver [76]. However, no positive relationships have been found with other ACEs, such as the quality of the relationship with the father, inconsistent discipline [70], or punitive discipline during childhood [66]. Miller et al. [57] found that eight of the twelve adversities analysed (other long-term parental separation, criminality, parental substance use disorder and mental illness, interparental violence, neglect, and physical and sexual abuse) were significantly associated with both perpetration and physical victimisation in dating relationships. Parental mental illness was the type of adversity most associated with perpetration (12.6%) and physical victimisation (10.2%) in dating. However, Nikulina et al. [76] found no association between parental mental illness or substance use and perpetration or victimisation in dating relationships.

#### 3.4.3. Emotional and Psychological Dating Violence and ACEs

##### Childhood Physical Abuse

Seven studies (21.8%) examined the association between childhood physical abuse and psychological dating violence. Although some studies found a positive association with perpetration [40,56], most did not [54,59,72,73,76]. In the case of victimisation, longitudinal [63] and cross-sectional analyses [40,56] found a positive relationship between childhood physical abuse and psychological dating victimisation. Other researchers, however, did not find the same relationship [73,76].

##### Childhood Psychological/Emotional Abuse

Eight studies (25%) investigated the association between childhood psychological/emotional abuse and psychological dating abuse. Fewer studies found a positive association with the perpetration of psychological and emotional dating violence [68,73,74] than those that found no association [40,54,59,76]. Of the studies that examined victimisation, most found that emotional abuse is associated with increased psychological abuse victimisation [40,63,73,74]. However, not all studies reported this association [76].

##### Childhood Sexual Abuse

Four studies (12.5%) considered the association between childhood sexual abuse and psychological dating violence. In relation to perpetration, Dardis et al. [59] found a positive relationship, but only among men. Other studies found no relationship for either perpetration or victimisation [40,54]. Nikulina et al. [76] found a positive relationship in their bivariate analyses for both perpetration and victimisation, but this relationship was not present in the multivariate models.

##### Witnessing Interparental Violence

Three studies (9.3%) examined the relationship between witnessing interparental violence and psychological dating violence. Two of these found no positive association with perpetration [56,72]. However, Nikulina et al. [76] did find such a relationship for both perpetration and victimisation. Gover et al.’s [56] results showed that father-to-mother violence was not significantly associated with psychological victimisation in dating relationships. However, mother-to-father violence was found to be significantly related to psychological dating victimisation among participants from South Korea, but not from the United States.

##### Parental Neglect

Five studies (15.6%) analysed childhood neglect and its relationship to dating violence. Longitudinal [63] and cross-sectional studies [74] found that emotional and physical neglect was associated with psychological perpetration or victimisation in both sexes. However, other studies did not find this same association for either perpetration [59,76] or victimisation [73].

##### Other Adverse Childhood Experiences

Only two studies (6.2%) investigated other forms of ACEs and their association with dating violence. These studies found that abuse experienced in dysfunctional homes is significantly associated with the perpetration of psychological violence [74]. However, Nikula et al. [76] found no relationship between experiences such as growing up in a household where one of the family members is incarcerated, living with a person with mental illness, or parental substance abuse and an increased risk of perpetration or victimisation.

#### 3.4.4. Sexual Dating Violence and ACEs

##### Childhood Physical Abuse

Two studies (6.2%) examined the relationship between childhood physical abuse and sexual dating violence. Neither study found a positive association between physical abuse and the perpetration of sexual violence [40,50]. However, Voith et al. [40] did find a positive association between childhood physical abuse and sexual dating victimisation, although only among men.

##### Childhood Psychological/Emotional Abuse

Three studies (9.3%) examined the relationship between childhood psychological abuse and sexual dating violence. Baller and Lewis [74] found a positive association with the perpetration of sexual violence, but the remaining studies did not [40,59]. In the case of victimisation, only one study analysed this relationship, finding no association between the two variables [40].

##### Childhood Sexual Abuse

Three studies (9.3%) examined the association between childhood sexual abuse and sexual violence. Dardis et al. [59] found that experiences of childhood sexual abuse were associated with perpetration of sexual violence for women, but not for men. Voith et al. [40] found that men who reported a history of childhood sexual abuse were more likely to perpetrate and experience sexual dating violence. However, Loh and Gidycz [52] found that while there was a significant relationship between the two variables, childhood sexual abuse was not predictive of perpetration of sexual violence.

##### Witnessing Interparental Violence

Only one of the reviewed studies (3.1%) analysed the relationship between exposure to interparental violence and dating sexual violence, finding no significant association [50].

##### Neglect

Of the two studies reviewed (6.2%), one found that physical neglect was associated with perpetration of sexual violence [74], while the other did not find the same relationship [59].

##### Other Adverse Childhood Experiences

A single study included other ACEs in relation to sexual dating violence, reporting that abuse experienced in dysfunctional homes is associated with greater perpetration of sexual violence [74].

#### 3.4.5. Cyber Dating Abuse and ACEs

##### Childhood Physical Abuse

Only one of the reviewed studies examined the relationship between childhood physical abuse and cyber dating abuse (3.1%), finding neither a direct nor an indirect relationship, through the mediation of early maladaptive schemas, between childhood physical abuse and cyber dating abuse [75].

##### Childhood Psychological/Emotional Abuse

Two studies (6.2%) looked at the relationship between childhood psychological abuse and cyber dating abuse. Celsi et al. [75] found that frequent experiences of emotional abuse during childhood were associated with an increased likelihood of perpetrating and experiencing cyber dating abuse. However, this was mediated through the internalisation of the emotional deprivation schema. Baller and Lewis [74] found moderate associations between childhood emotional abuse and cyber dating abuse victimisation.

##### Childhood Sexual Abuse

The search yielded no studies researching the association between childhood sexual abuse and cyber dating abuse perpetration or victimisation.

##### Witnessing Interparental Violence

Three studies (9.3%) examined the relationship between witnessing interparental violence during childhood and the perpetration and victimisation of cyber dating abuse. Cano-Gonzalez et al. [71] found that adults who had witnessed interparental violence were more likely to perpetrate psychological, sexual, and stalking cyber dating abuse. The association between interparental violence and cyber dating abuse occurred irrespective of the sex of the parent perpetrating or being subjected to violence. Similarly, Ramos et al. [77] found that adults who had witnessed more interparental violence reported higher levels of perpetration of cyber dating abuse. However, Celsi et al. [75] only found a positive and significant association for women who had witnessed interparental violence committed by mothers. This same study found no relationship between witnessing interparental violence and increased cyber dating abuse victimisation.

##### Parental Neglect

Two studies (6.2%) examined the relationship between parental neglect and cyber dating abuse. Celsi et al. [75] found that physical neglect was associated with the perpetration of pressure aggression and control monitoring. Childhood emotional, but not physical, neglect was associated with control monitoring, but only for women. However, Baller and Lewis [74] did find a positive association between childhood physical neglect and cyber dating abuse victimisation.

##### Other Adverse Childhood Experiences

Two studies (6.2%) analysed the relationship between other ACEs and cyber dating abuse. Baller and Lewis [74] found that abuse experienced in dysfunctional homes was associated with cyber dating abuse victimisation. Celsi et al. [75] examined early maladaptive schemas (specifically abandonment and emotional deprivation schemas) and their relationship to cyber dating abuse perpetration and victimisation. Their results showed the existence of a relationship between emotional deprivation schemas and the two forms of cyber dating abuse perpetration under examination: control monitoring and pressure aggression.

##### Combined Forms of Dating Violence and ACEs

Two studies (6.2%) investigated the relationship between ACEs and combined forms of dating violence. Voith et al. [40] found that men who reported a history of childhood sexual abuse were more likely to use polyperpetration against their dating partners. Abajobir et al. [63] found that the probability of experiencing various forms of dating abuse was higher for adults who were emotionally abused or neglected during childhood. Similarly, Voith et al. [40] found that physical abuse was related to polyvictimisation in adulthood, whereas childhood sexual abuse was not.

##### Other Forms of Dating Violence and ACEs

Adverse childhood experiences and their association with other forms of dating violence have not been widely researched. Only two studies (6.2%) included other forms of dating violence, namely harassment and threatening behaviour. Abajobir et al. [63] showed that experience of harassment was 1.63 times higher among those who had been emotionally abused as children. McClure and Parmenter [73] found that perpetration or victimisation involving threatening behaviour was related to childhood emotional abuse, emotional neglect, and physical neglect, although only perpetration of threatening behaviour was related to childhood physical abuse.

##### Sex Differences in the Relationship between Dating Violence and ACEs

Of the studies reviewed, 14 (43.7%) included sex-differentiated analyses of the association between ACEs and dating violence. Four of these found no significant differences based on the sex of the participants [54,67,72,73].

In relation to the perpetration of dating violence, Luthra and Gidycz [53] found that women with violent fathers were three times more likely to perpetrate dating violence. They did not find the same relationship among men. Similarly, Milletich et al. [55] found that childhood physical abuse was associated with the perpetration of physical violence among women but not men. Childhood emotional abuse was related to the perpetration of dating violence in men, but not in women. Dardis et al. [59] found that childhood sexual abuse is associated with sexual perpetration among women and psychological perpetration among men. Childhood experiences of maternal neglect were associated with physical perpetration in men.

The moderation and mediation models of perpetration also found differences depending on the sex of the participants. Loucks et al. [68] found that the association between childhood emotional abuse and the perpetration of psychological dating violence was statistically insignificant among women with relatively high levels of emotional communication skills. Lee et al. [61] found that among women, but not men, the relationship between childhood physical abuse and perpetration of physical dating violence was also mediated by perpetration of sibling violence and an anxious attachment style. Ramos et al. [77] found that higher perspective-taking and empathy buffered the relationship between parent-to-child violence and the perpetration of cyber dating violence among men more than among women.

With respect to victimization, Herbert et al. [24] showed that the risk of dating victimization for both sexes increases if adverse childhood experiences were experienced before the age of 16. However, they found that men who reported childhood sexual abuse, had witnessed domestic violence, or whose parents had separated were more likely to experience dating violence victimization than women who had experienced the same ACEs.

## 4. Discussion

Previous research has found that exposure to ACEs is associated with experiencing or perpetrating intimate partner violence in adulthood, such as domestic violence and abuse [16,79]. One explanation for this is that experiences in dysfunctional homes and experiencing or witnessing violence during childhood may be transmitted intergenerationally as similar behaviours occur or are endured in adult relationships [80]. Based on this premise, the aim of the present review was to examine the scientific literature in order to answer the following question: *what adverse childhood experiences within the family context are associated with dating violence in adulthood?* This scoping review includes 32 peer-reviewed articles from an initial sample of 599, from which the main information for each study has been extracted and summarized.

Adverse experiences that are associated with the perpetration of dating violence include: physical abuse [24,40,55,56,58,61,65,72,73,78]; emotional abuse [24,40,46,59,68,73,74,75]; sexual abuse [24,40,52,59,62,76]; witnessing family violence [55,69,72,77]; physical and emotional neglect [59,69,73,74,75,78]; low parental warmth [78]; parental mental illness, suicide attempt, criminal conviction, or separation [24]; dysfunctional homes [74]; the quality of the maternal relationship [65,78]; growing up with an incarcerated caregiver [76]; and early maladaptive schemas [75].

In the case of victimization, a significant association has been found with: physical abuse [24,40,51,53,55,56,58,63,64,70,73,74,76]; emotional abuse [24,40,63,73,74,75,76]; sexual abuse [24,39,40,54,60,64,67]; physical and emotional neglect [63,64,69,73,74]; witnessing interparental violence [24,51,55,56,69,76]; low parental warmth [58]; and parental substance abuse, mental illness, suicide attempts, criminal conviction, or separation [24].

The studies reviewed seem to suggest that childhood physical abuse is the adverse experience most associated with both perpetration and victimization of dating violence. This result may suggest that exposure to severe child abuse has the strongest association with dating violence, although it should also be noted that this is the experience most analysed in the existing studies (50%). However, as with all other forms of dating violence, the results are inconsistent. Moreover, the studies reviewed also seem to suggest that the results vary depending on the sex of the participants. The present review provides evidence that predictors of dating violence perpetration and victimization may be differently connected for men and women. However, not all studies analyse sex differences or include samples of both sexes, which also makes it difficult to draw clear conclusions.

Presented in this way, the existing evidence seems to lend some support to social learning theory and the intergenerational transmission of violence [81], which argues that, through learning processes, witnessing and experiencing different forms of violence in childhood leads to greater use of violence as an adult [50,65]. However, the findings of the present review are inconsistent and therefore this hypothesis cannot be confirmed. Contrary to expectations, the findings partially support the idea that those who experience adverse childhood experiences may be at greater risk of dating violence perpetration or victimization in adulthood. While there appears to be a link between adverse childhood experiences and dating violence victimization and perpetration, the underlying mechanisms of this relationship are not well understood. Moreover, the results of these studies are largely correlational rather than causal. In other words, it is not known whether experiencing childhood abuse leads to an increased likelihood of experiencing adult dating violence or, rather, if those who experience childhood physical abuse are also more likely to experience a variety of risk factors that increase their likelihood of being victims or perpetrators of violence in intimate partner relationships.

The inconsistency in the findings of the reviewed studies is in line with meta-analyses that have reported only the existence of a weak size effect on the relationship between experiencing and witnessing violence within the family and subsequent involvement in intimate partner violence [82]. Such mixed results have also been reported by methodological reviews of research into intimate partner violence. These indicate that the methodological variations between studies are so wide that their results are difficult to analyse because of contradictory findings [83].

### 4.1. Limitations and Recommendations for Future Research

There are several limitations to this scoping review. Firstly, search-related limitations such as the use of a limited number of databases and the search terms themselves may have led to the exclusion of some studies that addressed the question posed in this review. Similarly, the fact that only English language papers have been reviewed may mean that important studies published in other languages were omitted. Other search limitations relate to some of the decisions made in planning the search. Focusing on adulthood, looking at only one type of intimate partner violence (specifically dating violence), or considering only those ACEs occurring within the family context also limits the range of conclusions to be drawn from this review. Since dating violence occurs after childhood, future reviews should also explore dating violence in adolescence, and not restricting their associations to ACEs occurring within the family. Additionally, future reviews should extend the bibliographic search to other languages and other databases, and to reconsider the search terms in order to capture all the existing evidence and offer insights about what influences the relationships between ACEs and dating violence.

However, one strength of this scoping review is that the search has revealed some important limitations in the existing evidence on the question posed, which should be taken into account by future studies. In particular, the studies reviewed include a wide variety of methodological differences ranging from the definition of dating violence, the study design, the sampling techniques, the instruments used, and the statistical analyses performed. Furthermore, most of the studies reviewed included samples of university students, had cross-sectional designs, and the data collected from all 32 studies were self-reported. This makes it difficult to compare studies and draw conclusions about the impact of ACEs on dating violence victimization and perpetration. In fact, the results’ inconsistency may, at least in part, be due to limitations in the analytical approaches and the sample used. For instance, the studies are heavily based on retrospective reports of exposure to ACEs. Therefore, there has been no control for other individual, familial, and contextual factors mediating or moderating the relationship between ACEs and dating violence that could enhance understanding of such a complex phenomenon [63]. Future research should try to identify other factors that may play a role in mediating or moderating the relationship between ACEs and dating violence perpetration and victimization in adulthood, such as peer group relationships.

In terms of sample selection, most research has focused on university students and key risk factors may vary in populations with different characteristics [76]. Another important aspect of sample selection is that most studies were conducted in the USA with mainly Caucasian samples [76]. However, the cultural context may also lead to variations [56]. Other important methodological considerations that may influence the results are the measurement of the ACEs, the severity of the behaviours assessed and their duration. As Abajobir et al. [63] note, exposure to, for example, severe abuse in childhood, such as poly victimisation, is likely to lead to more frequent experiences of dating violence perpetration and victimization later in life. Therefore, as suggested by Ulloa et al. [39], it would be beneficial to explore the severity of the adverse experiences. In light of the results, it would be appropriate to replicate the studies carried out in order to unify the methodology and measures used [62] and conduct more cross-cultural studies. Additional research could also focus on comparing these associations across high, middle, and low-income countries. Further research is needed to obtain a complete picture of the relationships between ACEs and dating violence across different geographical locations, cultural contexts, and socio-economy status. Future studies should also be conducted to explore if the relationship between ACEs and dating violence is stronger when dating violence and adverse experiences within the family, but also in the social context, occur simultaneously. At the theoretical level, this review highlights the need for future research to start from a more concise conceptual model to define and measure ACES. A clearer model will allow the comparison of results among studies and will also offer a more precise understanding of ACEs’ health and behavioural correlates. In this sense, the conceptual model developed by Kalmakis and Chandler [38] can be very useful for research and trauma-informed care purposes.

### 4.2. Implications for Practice

In 1996, the 49th Assembly of the World Health Organization (WHO) declared violence prevention as a public health priority. Given the multicausal nature of this problem, the prevention of dating violence requires an understanding of different variables, such as individual, cultural and social determinants, including family and community factors. Several key public health implications arise from the evidence available on the relationship between ACEs and dating violence.

Although the present review has shown that ACEs are not always related to dating violence in adulthood, dating violence is likely to co-occur with other personal, family, and environmental problems that contribute to the underlying mechanism behind the cycle of violence. Intervention efforts must be directed to implement positive conflict resolution strategies in interpersonal relationships and address individual risk factors such as emotional dysregulation, alcohol consumption, emotional dependence, beliefs that justify violence and the myths of romantic love. Additionally, ACE prevention and trauma-informed care can have significant importance in preventing violence behaviours later in life. Preventing programs should adopt a life-course perspective that will require cross-sector interventions (involving health systems, schools, universities, social services, community organization and state security forces) addressing the economic, cultural and social contexts that facilitate the perpetration and victimization of aggressive behaviours. Among others, perinatal home visits, community-based programs providing parental skills and social support, young mentoring programs, behavioural health services and psychological therapy in primary care seem to be promising to overcome poor health status resulting from ACEs [84] and also could be helpful in preventing dating violence. Consequently, it is necessary to improve professional training of primary care and emergency physicians, paediatricians, school nurses, social workers and other public health practitioners for awareness, detection, and prevention of adverse childhood experiences.

Furthermore, it is necessary to detect and intervene dating violence early in life to avoid chronification. In line with previous research [57,75], intervention efforts with adolescents exposed to multiple childhood adversities could be useful to prevent the occurrence of dating violence and other forms of intimate partner violence in adulthood. Finally, considering that early experiences of childhood maltreatment may be related to dating abuse victimization and perpetration, we recommend examining and addressing early histories of trauma when intervening with victims and perpetrators in adulthood. Felitti and Anda [18] explain that the recognition of the trauma history can be therapeutic because it gives people the opportunity to reflect on the role that ACEs suffered have had on their lives and in the problem of subsequent violence.

## 5. Conclusions

This scoping review suggests that the relationship between ACEs and later perpetration and/or victimization is complex and may be mediated by other factors with which ACEs coexist, such as peer relationships, delinquency, and other risky behaviours [85,86]. Thus, while experiencing adverse childhood experiences is a factor associated with both perpetration and victimization of adult dating violence, it is likely to co-occur with other personal, family, and environmental problems that contribute to the underlying mechanism behind the cycle of violence. As suggested by Graves et al. [87], the effects of family violence and adverse experiences may lessen over time, and it may be that other concurrent factors become stronger predictors of dating violence. Therefore, adverse childhood experiences may not be a necessary or sufficient condition for experiencing dating violence.

## Figures and Tables

**Figure 1 behavsci-12-00162-f001:**
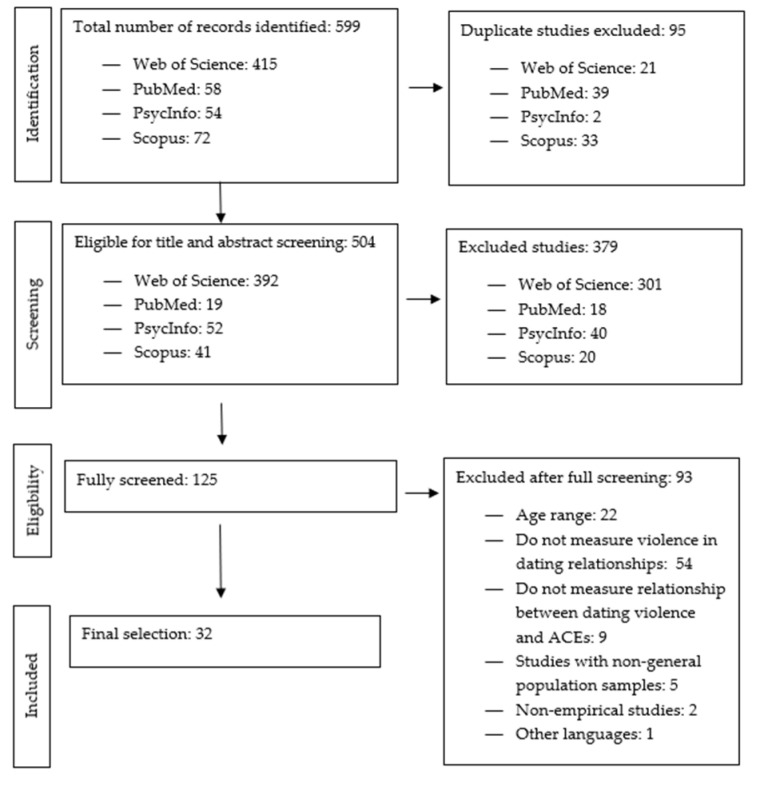
PRISMA flowchart on the identification, screening, and selection of articles for review.

**Table 1 behavsci-12-00162-t001:** Search terms used.

Search Terms
1. (“adult*” OR “young adult*” OR “emerging adult*” OR “early adult*” OR “18 yrs & older”).ti
2. (“dating violen*” OR “dating abus*” OR “dating aggress*” OR “cyber dating violen*” OR “cyber dating abus*” OR “cyber dating aggress*” OR “digital dating abus* OR “digital dating violen*” OR “digital dating aggress*” OR “electronic dating abus*” OR “electronic dating violen*” OR electronic dating aggress*” OR “intimate partner violen*” OR “intimate partner abus*”).ti
3. (victim* OR perpetrat* OR aggress*).ti
4. (“Adverse Childhood Experienc*” OR “ACEs” OR “advers*” OR “childhood neglect“ OR “childhood psychological abus*” OR “childhood sexual abus*” OR “childhood physical abus*” OR “exposure to substance abus*” OR “substance abus*” OR “exposure to mental illness” OR “parental mental illness” OR “mother treated violen*” OR “parental substance abus*” OR “criminal behavior in household” OR “sibling violen*” OR “family economic adversity”
5. (associat* OR correlat* OR mediat* OR moderat* OR determinant* OR predict*).ti
6. 1 and 2 and 3 and 4 and 5 search performed in each database.

**Table 2 behavsci-12-00162-t002:** Inclusion and exclusion criteria.

Inclusion Criteria	Exclusion Criteria
Participants aged 18 years or older.	Participants under 18 years of age.
Participants in the study had to belong to general populations.	Clinical samples or subgroups. For example, people with mental illness, federal sex offenders.
Quantitative empirical research, published in peer-reviewed journals.	Qualitative research, articles describing interventions or prevention and intervention programs, literature reviews, systematic reviews, conference papers, doctoral theses, journal articles.
Research investigating experiences of both in-person and online violence in dating relationships.	Research investigating both in-person and online forms of violence occurring outside of dating relationships such as among married or cohabiting couples, etc.
Research investigating at least one of the experiences linked to “adverse childhood experiences” within the family context.	Research not investigating at least one of the experiences linked to “adverse childhood experiences” within the family context.
Published in English.	Published in languages other than English.

**Table 3 behavsci-12-00162-t003:** Study rigour indices.

Study	Study Aim	Study Design	Study Selection	Selection Bias ^1^	Representative Sample	Stat Power	Response Rate	Measure Validity	Stat Sig	Confidence Interval	Quality Score
[50]	Yes	Yes	Yes	Yes	Unsure	No	Unsure	Yes	Yes	No	5
[51]	Yes	Yes	Yes	Yes	Unsure	No	Unsure	Yes	Yes	No	5
[52]	Yes	Yes	Yes	Yes	Unsure	No	Unsure	Yes	Yes	No	5
[53]	Yes	Yes	Yes	Yes	Unsure	No	Unsure	Yes	Yes	No	5
[54]	Yes	Yes	Yes	Yes	Unsure	No	Unsure	Yes	Yes	No	5
[39]	Yes	Yes	Yes	Yes	Unsure	No	Unsure	Yes	Yes	No	5
[55]	Yes	Yes	Yes	Yes	Unsure	No	Unsure	Yes	Yes	No	5
[56]	Yes	Yes	Yes	Yes	Unsure	No	Unsure	Yes	Yes	No	5
[57]	Yes	Yes	Yes	No	Yes	Yes	Unsure	Yes	Yes	Yes	9
[58]	Yes	Yes	Yes	No	Yes	Unsure	Yes	Yes	Yes	No	8
[59]	Yes	Yes	Yes	Yes	Unsure	No	Unsure	Yes	Yes	No	5
[60]	Yes	Yes	Yes	Yes	Unsure	No	Yes	Yes	Yes	Yes	7
[61]	Yes	Yes	Yes	Yes	Unsure	No	Unsure	Yes	Yes	Yes	6
[62]	Yes	Yes	Yes	Unsure	Yes	Unsure	Yes	Yes	Yes	Yes	8
[63]	Yes	Yes	Yes	Unsure	Yes	Unsure	Unsure	Yes	Yes	Yes	7
[64]	Yes	Yes	Yes	Unsure	Yes	Unsure	Yes	Yes	Yes	Yes	8
[65]	Yes	Yes	Yes	Yes	Unsure	No	Yes	Yes	Yes	No	6
[66]	Yes	Yes	Yes	Yes	Unsure	No	Unsure	Yes	Yes	No	5
[67]	Yes	Yes	Yes	Unsure	Yes	Unsure	Yes	Yes	Yes	No	7
[68]	Yes	Yes	Yes	Unsure	Unsure	No	Unsure	Yes	Yes	No	5
[69]	Yes	Yes	Yes	Unsure	Yes	Unsure	Yes	Yes	Yes	Yes	8
[70]	Yes	Yes	Yes	Yes	Unsure	No	Yes	Yes	Yes	No	6
[71]	Yes	Yes	Yes	Yes	Unsure	No	Unsure	Yes	Yes	Yes	6
[72]	Yes	Yes	Yes	Yes	Unsure	No	Unsure	Yes	Yes	No	5
[73]	Yes	Yes	Yes	Yes	Unsure	No	Unsure	Yes	Yes	No	5
[40]	Yes	Yes	Yes	Yes	Unsure	No	Unsure	Yes	Yes	No	5
[74]	Yes	Yes	Yes	Yes	Unsure	No	Unsure	Yes	Yes	No	5
[75]	Yes	Yes	Yes	Unsure	Unsure	No	Unsure	Yes	Yes	No	5
[24]	Yes	Yes	Yes	No	Yes	Yes	Unsure	Yes	Yes	Yes	9
[76]	Yes	Yes	Yes	Yes	Unsure	No	Unsure	Yes	Yes	Yes	6
[77]	Yes	Yes	Yes	Yes	Unsure	No	Unsure	Yes	Yes	No	5
[78]	Yes	Yes	Yes	Yes	Unsure	No	Unsure	Yes	Yes	No	5

^1^ Reversed score for the quality score.

## Data Availability

All data generated as part of this study are included in the article.

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
