# Peer review of "Associations between Adverse Childhood Experiences within the Family Context and In-Person and Online Dating Violence in Adulthood: A Scoping Review"

_behavsci, 2022, doi:10.3390/bs12060162_

Round 1

Reviewer 1 Report

Associations between Adverse Childhood Experiences within the Family Context and In-Person and Online Dating Violence in Adulthood: A Scoping Review

This is a relevant and important contribution to the readers of Behavioral Sciences. The article is well written, very well systematized, and scientifically sound. A few minor changes, though, would improve the overall quality of the article:

  1. Please clearly indicate the objectives of this review.
  2. Please include an implications sub-section, focusing on public health recommendations.

Best wishes.

Author Response

Thank you for your comments and the kind advice. We greatly appreciate your constructive comments that have helped us improve our paper. We have endeavored to incorporate the feedback and revised our manuscript accordingly. The itemized response is as follows:

  1. Please clearly indicate the objectives of this review. Authors' answer: the objective of the review has been indicated in page 3, lines 107-109.
  2. Please include an implications sub-section, focusing on public health recommendations. Authors' answer: following you advice a sub-section regarding implications for practice has been included in the discussion section. Page 8. Thank you.

Reviewer 2 Report

First of all, congratulations for investigating such a necessary topic as violence. Also for the time it takes to carry out this work, hence my congratulations for the time and effort.

The article entitled "Associations betwween adverse childhood experiences within the family context and in-person and online dating violence in adulthood: a scoping review", presents a review according to PRISMA guidelines.

The objective is a review of adverse childhood experiences within the family context and in-person and online dating violence in adulthood.

The introduction conceptualises childhood issues well and relates them to later dating violence in adulthood.

The design makes it clear that it has followed an Arksey methodological design and that it has followed PRISMA. In addition, it presents the search terms, which is welcome. It also details the reasons for inclusion and exclusion for their study, and presents the PRISMA flow chart.

It also discusses the methodological differences in the studies, the different assessment instruments used and provides a detailed analysis of the violent factors occurring in childhood that are related to the development of violence in adulthood.

In the discussion, the authors present a discussion of the findings, and express the limitations of their review related to the search.

I congratulate them on their work, my comments of suggestions for improvement are:
-Incorporate at the end of the discussion theoretical implications that this work has for the review of researchers at a theoretical level.
-Incorporate at the end of the discussion practical implications for society and the researched sector.
-Explain the strength of this review.
-To put forward a future line of research to be developed.

Best regards, 

Author Response

Thank you for your comments and the kind advice. We greatly appreciate your constructive comments that have helped us improve our paper. We have endeavored to incorporate the feedback and revised our manuscript accordingly. The itemized response is as follows:

-Incorporate at the end of the discussion theoretical implications that this work has for the review of researchers at a theoretical level. Authors' answer: thank you for this suggestion. We have incorporate implications at the theoretical level in the discussion. Please, see the paragraph in page 8, lines 769-775.

-Incorporate at the end of the discussion practical implications for society and the researched sector. Authors' answer: Thank you. A new subsection regarding practical implications have been added in page 8.

-Explain the strength of this review. Authors' answer: The strength of the review has been indicated in page 7, line 735.

-To put forward a future line of research to be developed. Authors' answer: thank you for this suggestion. Future lines of research have been further developed in page 8, lines 763-769.

Reviewer 3 Report

I have reviewed this manuscript, which presents a very precise question that translates from the title: the association between adverse childhood experiences and dating violence in adulthood. This review article examines the scientific literature to answer that specific question. Therefore, accuracy is one of the strong points of the text.

This review article is thoroughly researched and rendered with precision. It presents valuable (but insufficient) information about the matter.

Another strong point of the manuscript is the honesty in the treatment and discussion of data, the limitations and recommendations and conclusions.

The search strategy is correct, although there is a certain gap: those years that clearly do not belong to childhood and in which cases of dating violence can already be found. The authors have demonstrated their intention to establish robust selection criteria for bibliographic documents, including those that looked at dating violence among participants over the age of 18. However, given the limited number of resulting bibliographic sources (32 of the 599 records identified), for this enormous amount of work to bear more fruit, a revision that extends its limits may be necessary, since in my view (and it seems that in that of the authors), the limits of the investigation may have turned out to be limitations. In fact, we find that this review article devotes more text to Limitations and recommendations than to Conclusions. For this reason, I believe that this article lacks useful and rich conclusions, without undermining the effort made.

In general, the question is very precise (even without being geographically constrained). The results give a review of few included articles based on the inclusion or exclusion criteria. I wonder if the authors have found sources that specify cases where dating violence and childhood violence occur simultaneously.

I do acknowledge that one of the findings is remarkable: evidence that predictors of dating violence perpetration and victimization may be differently connected for men and women. And yet, as some studies do not include samples of both sexes, the authors question the possible conclusions. Other aspects, also recognized by the authors, such as the methodological difference used in different studies, lead to contradictory findings.

I think that the question formulated for the review is interesting and can be very useful for society. But among the limitations pointed out by the authors, I want to highlight those that I think that would help make their findings more complete:

- Extension of the bibliographic search to other languages, to other databases and reconsideration of the search terms. It must be taken into account that a fairly universal question was asked, and yet the results seem restricted to the case of the USA (which was not the initial objective).

-For the usefulness of the review, the decisions made in planning could be extended. Since dating violence occurs after childhood, perhaps expanding the experiences of violence in that period of life, and not restricting it to ACEs occurring within the family, could offer more useful results.

In another vein, The tables are concise and easy to follow.

As the same authors deduce in light of the results, it would be appropriate to replicate the studies carried out in order to unify the methodology and measures used and conduct more cross-cultural studies. However, I believe that this revision is correct, it follows the required steps rigorously, it is well structured, and the sources are used neatly. I would wait for a subsequent research article to give more soundness to the work already done, but there is no doubt that much progress has been made in a process that still requires some adjustments in decision-making. The remaining examples are so few that they should be completed with a broader perspective. In short, it is correct for what is stated, but underdeveloped. The topic is very relevant, and the authors, (by asking themselves the question) have identified that their answers can lead to findings in their field. Therefore, they have identified a gap in knowledge.

Author Response

We thank you the reviewer for the time and effort  put into reviewing our manuscript. All your comments have enabled us to improve our work and we are agree with the critics arise on previous research and the limitation of our own review. Specifically we have attended your comments incorporating lines for future research on page 8, lines, 763-769. We have also further remarked the limitations expressed on your comments (page 7, lines 728-733) including a call for future reviews, and also indicating some implication of our findings in page 8 (lines 778-812)  under a new subection regarding implications for practice.